# Health Deficits Among People Experiencing Homelessness in an Australian Capital City: An Observational Study

**DOI:** 10.3390/ijerph22020135

**Published:** 2025-01-21

**Authors:** Susan J. Gordon, Nicky Baker, Tania S. Marin, Margie Steffens

**Affiliations:** 1College of Nursing and Health Sciences, Flinders University, Bedford Park, Adelaide 5042, Australia; nicky.baker@flinders.edu.au (N.B.); tania.marin@flinders.edu.au (T.S.M.); 2Adelaide Dental School, The University of Adelaide, Adelaide 5000, Australia; margie.steffens@adelaide.edu.au

**Keywords:** health status, accelerated ageing, people experiencing homelessness, vulnerable population, public health

## Abstract

People experiencing, or at risk of, homelessness face challenges that result in poorer health outcomes compared to those in stable housing. This study provides the results of over 40 health measures that capture the health status of a group of people in temporary accommodation due to experiencing homelessness, aged 22 to 84 years, in an inner-city location, invited to participate in a comprehensive assessment of physical and psychological health. Evidence of accelerated ageing was found, with 44.2% of people being clinically frail, 63% having poor functional movement, and 36% having pain associated with oral health. Additionally, 90.6% of participants showed health risks due to nutritional deficiencies, over half reported poor sleep quality, 55.3% reported having psychological distress, and almost half reported fair or poor overall dental health. This study suggests a pathway to providing a relatively easily implemented series of health assessments to help respond to a group of underlying causes for accelerated ageing among a group of inner-city people experiencing homelessness. This work can be used to inform the prioritisation and development of community-based health services to address functional deficits that may contribute to accelerated ageing.

## 1. Introduction

People experiencing homelessness (PEH) face a myriad of challenges to achieve similar health outcomes to those in stable housing [1,2,3,4]. Research has shown that lived experience of homelessness is associated with poorer physical and mental health outcomes [5,6], significant chronic and infectious disease morbidities [4,7], large decreases in life expectancy [1,5], greater rates of falling [5], and significantly higher use of acute care services, such as emergency departments (EDs) [8,9]. The experience of homelessness is also shown to negatively affect longer-term health, whether an individual remains homeless or not [10], with those exposed to long periods of homelessness showing even poorer health [4,11]. In Australia, those reporting being homeless are among the most socially and economically disadvantaged [12,13], and it has been shown that they may die up to 33 years earlier than those in secure housing [14].

Homelessness is a significant and complex biopsychosocial issue affecting persons of all ages and backgrounds [1,3], and there is a growing homelessness problem among adults globally [2], with housing conditions recognised as an important determinant of health identified in the Sustainable Development Goals [15]. However, definitions of homelessness vary across countries, which challenges comprehensive global assessment of its prevalence and consequences [1,2,5]. In Australia, a person experiencing homelessness is generally defined as any person living in a dwelling that is inadequate, has no tenure (or with short and non-extendable tenure), who does not have control of, and/or access to, space for social relations, and has no suitable alternative accommodation available to them [16,17], with those living in ‘marginal housing’ considered to be at risk of homelessness [18]. Homelessness has been named as a national priority for federal and state governments in Australia [19], with governments (state, territory, and federal) committing to a systems approach, driven by prevention and early-intervention strategies to address the main predictors that lead to people experiencing homelessness. These are: limited access to safe and appropriate housing that is affordable; incomes below those that support adequate tenancies; domestic and family violence; and poorly resourced support services [20].

One in seven (15.8%) PEH in Australia are aged 55 years or over [19]. As older people live with more complex health problems, including disabilities, chronic diseases, and geriatric symptoms [21], this adds a further layer of disadvantage to this demographic who may find it more challenging to access the specialist health services they require, both economically and socially, for best-practice preventive or chronic disease management [22,23]. Consequently, their health management is often crisis based, ‘managed’ by frequent unplanned attendances at after-hours medical care and hospital EDs which can result in potentially avoidable ward admissions [8,9,24], adding to the mortality gap between those housed and those experiencing homelessness due to inadequate healthcare [6,25].

‘Accelerated ageing’, where people experience early onset of conditions usually associated with older age, is seen among PEH globally at younger ages [13,26]. Two recent reviews [5,13] have shown that PEH can experience geriatric conditions and symptoms of frailty at younger ages (those aged 40 to 50 years). This accelerated ageing affects both PEH for the first time, those finding themselves at risk of homelessness, or those having had experienced homelessness for a while [27]. This may be due to their inability to find stable housing and, thus, not being exposed to the same supports that come with ‘ageing in place’ [28], or psychosocial determinants such as: increasing mental health issues, combined with various addictions, including gambling; inability to access social welfare and support programmes; financial insecurity; and/or a lack of social connectedness [2]. A systematic review using administrative data from six high-income countries, despite noting limitations due to paucity of the data and probable lesser completeness of data for homeless populations, stressed that it is imperative we begin to better understand the consequences of homelessness, by suggesting the predictors are key for a “development of policy or individual interventions to reduce homelessness and its adverse effects” ([1], p. 1618). Addressing these structural drivers of homelessness may be achieved more readily by partnering with organisations that work with people at risk of, or experiencing, homelessness and supporting them by providing a kind of transient stability to allow for health improvement initiatives to occur. This paper explores this by partnering with a community housing project, Common Ground [29], in Adelaide, South Australia (SA). Common Ground supports PEH as well as those at risk of homelessness. Their model of care is based on the principles of Housing First [30] and offers services and accommodation for PEH (as per the Australian Bureau of Statistics [ABS] definition [17]) ranging from rough sleepers or those staying temporarily with friends or relatives, through to those who previously lived in overcrowded, temporary, or inadequate dwellings. At the time of this study, Common Ground provided affordable accommodation through the National Housing and Homelessness Authority Agreement and Housing Choices SA [31]. In addition, tenants were offered dental and limited medical services (including vaccination and screening).

Although we cannot ‘measure’ accelerated ageing directly, we provide here the outcomes of a core set of health assessments able to measure functional decline in PEH and compare them by age and gender, and to expected normal values where they are available. To our knowledge, this number of health assessments has not been undertaken with a homeless population in this type of study before. This evidence may provide guidance for the prioritisation and development of community-based health services for PEH.

## 2. Materials and Methods

This cross-sectional, observational study describes the outcomes and consequences of a comprehensive set of health assessments involving PEH in inner Adelaide, SA, previously described as appropriate and acceptable for use in this population [32]. To overcome the lack of a comprehensive register of PEH in this capital city, convenience sampling with snowballing was used [33]. Recruitment occurred by direct invitation from staff of Common Ground, or by indirect invitation through peers, or other people seeing posters when they attended the organisation or any of its service network sites. Participants were offered appointments over a six-day period, and on the day of assessment, were asked to give written consent before completing a standard physiological risk assessment to establish fitness for study participation.

Informed consent was obtained from all participants, and ethical approval was provided from the Southern Adelaide Clinical Human Research Ethics Committee (SAC HREC): reference 222.17. Methods were carried out in accordance with the relevant guidelines and regulations of SAC HREC.

### 2.1. Study Participants

Participants were eligible for inclusion if they were aged 18 years or older and met the ABS definition of homelessness: any person living in a dwelling that is inadequate, has no tenure (or with short and non-extendable tenure), who does not have control of, and/or access to, space for social relations, and has no suitable alternative accommodation available to them [17]. Anyone who presented with pain on the day or had results for any of the screened physiological variables that were outside of normal values (blood pressure, blood oxygenation, heart rate, respiratory rate, and temperature) were excluded from study participation to minimise risk to the participant. All people attending their appointment were invited to a shared lunch and provided with a selection of oral and personal hygiene products, regardless of whether they participated in the study.

### 2.2. Health Assessments

Over 40 health measures were included to capture information on the general physical and mental health of the individual. These measures were informed by a systematic literature review of health screening and assessment tools suitable for PEH [34], previous physical testing of a housed community dwelling population living in the same city [35], and consultation with staff and health professionals working at the partner organisation, Common Ground, and its networks, to identify any changes in administration likely to be needed when working with PEH. To ensure all constructs were relevant to this population, before being implemented, some alterations were made to language which is not always relevant to PEH (e.g., not to assume that PEH have access to food preparation and storage services, as well as bathroom and toilet facilities). The validated surveys and objective measurements were divided into eight stations and participant results were recorded in a booklet.

Assessments were delivered across eight stations staffed by appropriate health professionals: a social worker for cognition and psychological distress; audiologists for audiometry; physiotherapists trained in spirometry for lung function, functional movement screen, grip strength, response time, dexterity, and walking tests; dietitians for anthropometry; general practitioners (GPs) for skin, eye, and foot health; and oral hygienists and dentists for oral health. Each appointment was 150 min, and at each station, the measurement was described in detail to the participant at the time of testing, and verbal consent was asked for prior to participation. Participants could decline to participate in any station or measurement, and those considered at risk of harm from an individual measurement were excluded from participating but were able to move on to others.

### 2.3. Data Collection and Managment

Data collection consisted of a mix of self-report using validated questionnaires provided on paper and objective measurements made by the qualified health professional, each of which was an indicator of accelerated ageing. Health assessment data presented in this analysis include anthropometry (body mass index [BMI]: weight/height^2^, waist circumference, hip circumference, and muscle mass); ear health (self-report measured by the Speech, Spatial and Qualities of Hearing Questionnaire [SSQ5] [36], audiometry using the Functional Hearing Assessment [37], and physical inspection); cognition assessed as the General Practitioner Assessment of Cognition (GPCog) [38]; dexterity (Purdue Dexterity Test) [39]; foot sensation (monofilament testing) [40]; clinical frailty (Clinical Frailty Scale) [41] and oral frailty [42]; diet and nutrition, including the DETERMINE nutritional health questionnaire [43]; functional strength and stability/flexibility (Functional Movement Screen [FMS]) [44]; history of health service use in the past six months; lung function (FEV1, FVC, ratio) [45]; mobility (Six-Minute Walk Test [SMWT]) [46]; oral health (the Australian Research Centre for Population Oral Health [ARCPOH] oral health questionnaire [47] and the Oral Health Impact Profile questionnaire [OHIP-14] [48]); pelvic floor bother [49]; protective health behaviours (screening and vaccination); psychological distress (Kessler Psychological Distress Scale [K10]) [50]; skin condition; sleep quality (Pittsburgh Sleep Quality Index [PSQI]) [51]; and grip strength (age- and gender-adjusted) measured using a handheld dynamometer [52,53,54];. The assessment descriptions have been previously reported [55]. Total scores were calculated for each of the validated measurement tools using the published scoring methods [36,38,41,43,47,48,50,51]. Predicted objective scores were calculated for forced expiratory volume at one second (FEV1) and forced vital capacity (FVC); lung ratio was calculated as FEV1/FVC [56] and reported as having obstructive, restrictive, or mixed-pattern airflow.

### 2.4. Data Analysis

Analyses were conducted using IBM SPSS Statistics version 29.0.2 (Clarivate, Chicago, IL, USA), R (R version 4.4.1), R studio [57], and Microsoft Excel. Data were investigated for normality and all variables met assumptions for the statistical tests used [58]. Categorical data are reported as numbers and percentages with 95% confidence intervals (CIs); continuous data as mean (M) with standard variation (SD). For assessments where recommended thresholds were available, these variables are recoded and reported as ‘at risk’ or ‘not at risk’, and differences between gender and age groups tested using a chi-squared (ꭕ^2^) test of independence. Where more than 20% of cells were determined to be ≥5, Fisher’s Exact Test was used [59]; for all other cases, *p* values are reported as the Pearson chi-square value. Where cut-offs were not published, data were analysed as continuous. These scale variables were checked for normality, and for those not meeting the assumptions due to outliers, bootstrapping was employed. All other variables not meeting assumptions for normality were recoded as categorical. Differences for all scale variables were tested using t-tests and one-way analysis of variance (ANOVA).

## 3. Results

An estimated 120 PEH heard about this study from posters, direct invitation by Common Ground staff, and word of mouth. A total of 90 (75%) indicated interest in participating, and 70 consented to risk screening. Pre-assessment risk screening for heart rate, body temperature, blood pressure, respiratory rate, and blood oxygenation measures were within expected ranges; therefore, no one was excluded due to risk of adverse health events. All 70 were invited to participate in testing. Of these, 17 people declined (24.3%), and 53 consented to at least one testing station. Rates of completion, refusal, and non-completion have been previously reported [32].

Participants ranged in age from 22 to 84 years (M 49.1, SD 14.9), with 58.5% (*n* = 31/53) aged under 55 years (see Appendix A). Thirty (56.6%) were men, and overall, women were younger than men, but not significantly (women range 22–69 years, mean 46.1, SD 14.3; men range 23–84 years, mean 51.4, SD 15.2). Over half of participants reported having high school-level education (54.7%), 20.8% had a post-high school certificate or trade, and most reported that their income came from a pension (90.6%). Participants were more likely to be single (including those divorced, separated, or widowed), with men statistically significantly (*p* = 0.034) more likely to be single when compared to women; almost all (88.7%) spoke English as their primary language. Over 90% of participants lived in the temporary accommodation provided by Common Ground (92%) with the remainder living in shared, short-term rentals (8%), fulfilling the criteria used to define homelessness, e.g., having no tenure or short and non-extendable tenure [16]. All had previously experienced living in a dwelling that was inadequate or did not allow them to have control of, and access to, space for social relations before coming to Common Ground. All participants attempted at least one health measurement, and participant engagement with the self-report questionnaires led by the station health professional was high. The main finding (see Table 1) was the moderate to high risk of poor nutritional health among participants (no difference by gender or age group) as measured using the DETERMINE questionnaire (90.6%); the most common contributors to the increased nutritional risk were eating alone most of the time (60.4%), taking three or more prescription drugs each day (45.3%), having an illness that had altered the kind and/or amount of foods eaten (41.5%), eating fewer than two meals a day (41.5%), and eating few fruits or vegetables or milk products (39.6%). Almost one in five (38.0%) participants also reported that their water consumption was below recommended (less than 2.1 L per day for women and 2.6 L per day for men) [60], their sugary drink intake was more than half a litre per day (32.0%), and self-reported rates of junk food consumption were more than twice a week for nearly a third of participants (28.8%). Over a third of participants were obese (36.4%), two-thirds measured over the recommendation of waist circumference for their gender (64.7%), and 42.1% were under the recommended age- and gender-adjusted muscle mass percentage.

Over half (56%) of participants also reported poor sleep quality on the PSQI (those in the younger age group were significantly more likely to do so), with 47.7% saying their sleep was broken three or more times a week due to having to get up in the night to use the bathroom, 40.9% using sleeping medication three or more times a week, 38.3% not being able to get to sleep within 30 min at least three times a week, and 25.5% sleeping less than six hours a night; one-third of participants (34%) rated their sleep quality as fairly or very bad. Additionally, 44.2% were clinically vulnerable or mild to moderately frail, and 55.3% of participants were found to have psychological distress, which was significantly higher among females and those who were younger

Almost half (47.7%; *n* = 21/44) of the participants who supplied data for the ARCPOH oral health questionnaire and the OHIP-14 rated their overall dental health as fair or poor, with a third of participants (31.8%) reporting they thought their dental health was a little/much worse than their overall health. Over half of participants (59.1%) reported being affected to some degree (fairly/very often or always) by at least one of the seven domains, with those 55 years and over significantly more likely to be affected. Just over 20% in both older and younger age groups reported having false teeth, and 80% of participants had had teeth extracted in the past (100% of those aged 55 years of over). One-third (32.4%) had less than 20 functioning teeth (significantly more likely for those aged 55 years or over—52.9%), and when asked if they would see a dentist, 52% said that they avoided seeing a dentist due to cost and 58% (69% of younger PEH and 40% of those who were older) said that they would find it very hard to pay, or could not pay, an AUD 100 dentist bill.

However, for other preventative health behaviours, all had undergone either blood, bowel, cervix, or prostate screening, and just under half had been vaccinated against flu, shingles, meningococcal, and/or pneumonia in the past year. Participants were also examined for nerve damage (peripheral neuropathy) using the monofilament foot test which showed that 13% of participants had abnormal foot sensation, a possible indicator of conditions such as diabetes [61]. On further skin examination, 38% (*n* = 20) had some sort of skin rash (on the face and back most commonly) due to seborrheic dermatitis, rosacea, and/or psoriasis, and 61% (*n* = 32) showed evidence of scarring.

Women were significantly more likely to have psychological distress, compared to men, and men were significantly more likely to have a waist-to-hip ratio over the recommend 90%, an unknown hearing problem when tested, or experienced food insecurity in the past week, compared to women. Although CIs are wide (see Table 1 and Appendix A), due in part to having a small sample and therefore not all differences being statistically significant, the following highlights where the main clinical differences were found. Those 55 years or over were significantly more likely to have the following risks identified:Measure over the recommended waist circumference.Be clinically frail.Have an unknown hearing problem and have wax/wax occlusion/pus or discharge in one or both ears.Show obstructive/restrictive or mixed airflow pattern in the lung function test.Be at risk for poor functional movement in the FMS.Have presented at an ED in the past 12 months or had a fall, or near fall, in the past 6 months.Have experienced food insecurity and reported having a poor appetite in the past week.Show that their oral health was affected by scoring one or more on at least one of the OHIP-14 domains.

## 4. Discussion

This study has identified aspects of health that contribute to overall poorer health outcomes across multiple performance domains and body systems for PEH. This contributes to the understanding of how the experience of homelessness may compromise both physical and psychological health [62]. Many of the identified poorer health outcomes, across multiple domains, are associated with chronic health conditions that, cumulatively, are shown to be associated with mortality and morbidity among PEH [1]. Further, the findings suggest an opportunity for assessments and associated interventions to be a priority for access via organisations providing temporary accommodation or other services to PEH to improve health status. A significantly high percentage of PEH in this study were found to have compromised cognition and memory across gender and age groups, indicating possible cognitive impairment that warrants a follow-up assessment in the next 12 months. This finding is supported by an umbrella review of cognitive functioning in homeless adults which reported up to 55% of PEH having cognitive deficiency [63]. Cognitive assessments in this study found 60% of PEH returning below-threshold scores. The prevalence of cognitive impairment in the Australian population is currently unknown, but an international review study has estimated the standardised prevalence of mild cognitive impairment in adults aged 60 years and older to be between 6% and 12% [64], far lower than found in this study. This may in part be attributed to the tools used to calculate differences in the criteria of mild cognitive decline [64], or a true reflection of SA’s population of PEH.

There are multiple opportunities to address poor health outcomes and the resultant incipient frailty that has been associated with accelerated ageing among PEH. Frailty rates in our sample were four times higher (44%) than SA population norms for pre-frailty/frailty (9–12%) [65]. In this study, we also found a significantly higher prevalence of poor functional movement among those aged 55 years and over, but also a high prevalence (63%) of those failing three of more of the functional movement assessments used to assess strength and stability (FMS deep squat, hurdle step, lunge, knee or toe push up, rotary stability, flexibility shoulder range, and straight leg raise). This was alongside high levels of those experiencing pain, at least one chronic health condition, and sleep problems. These findings are supported by a scoping review of physical functioning of people experiencing homelessness that found mobility impairments were more common in homeless populations because of pain, chronic health conditions, sleep problems, fatigue, and early onset of age-related conditions [66]. Additionally, a third of the study cohort of PEH reported diminished foot sensation, commonly associated with poor balance, falls, unexplained foot injury, and poor mobility due to poor foot hygiene and improper footwear [56].

The poor levels of sleep quality identified in this study have been reported as inevitable for PEH due to their environmental context in other cohorts [67]. A study of 2144 healthy people aged between 43 and 71 years in Spain found approximately 40% reported poor sleep quality using the same instrument (the PSQI) as used in this study [68]. However, a systematic review of sleep quality in adults in lower- and middle-income countries found significant variability in prevalence which could not be explained by method of assessment or location [69]. Poor sleep quality has been reported in approximately four out of five people with poor mental health [70], and findings from our study concur with the strong coexistence of homelessness, poor sleep quality, and poor mental health.

Recent investigations of poor mental health using the K10 have identified high, or very high, distress in 11% of Australian adults, with the most socially and economically disadvantaged populations reporting distress rates at more than twice the national average (27%) [65]. These rates are still much lower than the proportion of PEH reporting elevated distress (55%) in this study. Results here also show that 75% of women, and 62% of those in the younger age group, were significantly more likely to be experiencing psychological distress. Understanding mental health conditions in this population is complex, where mental health may contribute to, or be a result of, homelessness [71]. A systematic review of interventions for women who are experiencing homelessness that target these high levels of poor mental health has detailed some recommended practices moving forward [72]. However, most studies highlight high levels of poor mental health due to the added responsibilities women face in life, e.g., caring responsibilities for children, leaving a gap to explain the increased psychological distress among younger women in this study.

The latest AIHW data show that two in three (67%) Australian adults are overweight or obese, which rises to between 75% and 80% in those 55 years and over [73]. In our study, we found much lower rates of obesity (44%) and overweight (15%), with those in the lower age bracket more likely to be obese, and significantly more likely to be overweight (*p* = 0.016), with more women than men falling into this category. This is supported by the younger ages (those less than 55 years) being significantly more likely to not have a healthy weight overall (79%; *p* = 0.007) compared to those older (40%). However, those who were older or male were more likely to be at risk for having a waist-to-hip ratio over the recommended ratio of 90% for men and 85% for women. We also found high levels of food insecurity (42%) and very high levels of nutritional risk (90%). The mechanisms between food insecurity and BMI are complex and there is limited literature examining this phenomenon among PEH [74]. In a Boston (USA) study [74], similar proportions of overweight or obese people in community dwellings and homeless populations were found, leading the authors to attribute this to the hunger–obesity paradox, where poor individuals eat cheap, energy-dense foods, resulting in hunger and obesity occurring simultaneously in the same person. The Boston cohort reported a much lower prevalence of underweight people (1.6%) than the PEH in this study (13.1%), which may be explained by our small numbers. Further investigation of this is warranted in future studies.

A healthy diet reflects appropriate intake of nutrients and has been shown to protect against chronic disease, poor health, and premature ageing [75,76]. Nutrition should not be considered in isolation from oral and general health, and its relationship to weight and function is synergistic. Reasons for poor eating behaviours may include physiological changes related to ageing and ill health, such as impaired smell and taste, becoming satiated more quickly and therefore not eating as much, living alone and/or having minimal social contacts, which may lead to not being motivated to prepare adequate meals, experiencing dental problems, having poor exercise behaviours leading to being less hungry and thirsty, and taking multiple medications [77]. Other reasons why nutritional intake differs between people, including culture and race, income, lack of nutritional literacy, and traditional eating patterns, apply to both housed and homeless populations; however, food insecurity, poverty, lack of access to nutritious foods, and limited capacity to prepare, cook, or store food are particularly relevant to PEH [78]. In this study, we found good levels of water consumption (1.5–2 L per day), although still below Australian recommendations [60], low sugary drink intake (less than half a litre per day), except among older people, where this rose to nearly a litre per day, and self-reported rates of junk food consumption of once or twice a week, in line with Australian dietary guidelines that suggest a small amount of junk or discretionary food can be included in a healthy, balanced diet [79]. Again, margin of error was large due to small numbers, and it is important to interpret these results carefully as they are all self-reported recall.

Oral health has been shown to be linked with frailty [42], highlighting the importance of interventions to improve oral health as we age [80]. It has been shown that having 20 functioning teeth or less increases the rate of oral frailty by having a direct impact on the consumption of fruit and vegetables, proteins, and micronutrients, as it affects the ability to chew [42,80]. We find here that in this population of PEH, the average number of tooth extractions was almost 11, with some individuals having had up to all 32 teeth extracted (M 10.8, SD 11.1). Using the OHIP-14—originally developed for elderly people but found to be useful in assessing quality of life for dental needs and therefore used in this study [81,82]—we provide further evidence of poor oral health among this cohort of PEH. Significant numbers of PEH reported physical pain, psychological discomfort and physical disability (both 36%), and psychological disability (32%) due to their oral health, compared with Australian population norms of 10%, 8.5%, 2.5%, and 2.4%, respectively [83]. A systematic review has presented an examination of strategies available to PEH, showing that oral healthcare is generally only accessed in an emergency, despite the established and concerning evidence that poor oral health negatively affects overall health and wellbeing [84].

When comparing this to published norms from an AIHW analysis of the social impacts of oral conditions on quality of life among a representative sample of Australians [83], proportions of PEH in this study were much more likely to experience at least one impact (fairly often or very often) on all seven domains of the OHIP-14 (20.5%, 36.4%, 36.4%, 36.4%, 31.8%, 18.2%, and 29.5%) in the preceding year, compared to people not experiencing homelessness (9.9%, 10.6%, 8.5%, 2.5%, 1.8%, and 5.9%). The evidence here that PEH are also avoiding seeing a dentist due to cost or fear highlights an urgent need for the provision of safe and appropriate oral health services for this population. With a recent publication discussing the inaccessibility to care and strategies to improve outcomes among PEH in Australia, there are some encouraging statements regarding sustainable services that may improve overall oral and general health outcomes [85].

Testing for hearing level revealed that almost half (45.3%) of PEH reported below-threshold scores for the American Speech–Hearing–Language Association (ASHA) audiometry test score, with a mean score of 6.1 (SD 2.3); additionally, 10% reported hearing-related disability. Blackwell et al. [86] suggest that, worldwide, between 2% and 8% of adults over 45 years suffer some form of hearing-related disability, in line with the reported rates in this study. This is supported by a high number of people with ear wax, wax occlusion, pus, or discharge during the ear health inspection (38%), which may lead to the comparable proportions of people with unknown hearing loss (34%), which was significantly greater (*p* = 0.006) in men (48%) and those aged 55 years and over (50%). This is a health complaint that can be easily treated in a day clinic, alongside free hearing checks for PEH.

It is well established in the literature that older adults experiencing homelessness have more complex health issues and social challenges, such as access to health and accommodation services, than those in stable housing, or those who are younger [27]. This study provides further evidence for this in an Australian setting. Examples of bringing community service organisations together with homeless populations to provide services that are needed most are not unique to Australia [87]; however, accommodation linked to these services appears to provide a more comprehensive solution in Australia [29]. Although numbers of PEH in supportive accommodation in Australia has remained stable over the past two decades (19.3% in 2006; 20.7% in 2011; 18.2% in 2016; 19.8% in 2021) [19], supportive accommodation for PEH has been shown to help reduce inappropriate health service use [88], and in the case of Common Ground, they offer health-enhancing services such as oral health, nutritional advice, and mental health services to clients who had previously lacked any healthcare support, resulting in socio-emotional benefits to clients [89]. All participants in this study have some community connectedness due to their association with Common Ground which gives them greater access to health services. This was evident in the relatively high proportion of participants having undergone screening and the high rates of vaccination against flu, shingles, meningococcal, and/or pneumonia in the past year (50%); this study was conducted before the severe acute respiratory coronavirus 2 (SARS-CoV-2) pandemic. In comparison, Australian inner-city rates of vaccination for those 65 years and over against influenza and pneumococcal disease sit at 50% [90].

Initiatives to support PEH should target issues that are likely to have the biggest impacts across multiple health domains [22]. The participants in this study had already accessed limited medical and dental services through the partner agency Common Ground. The study findings suggest that this organisation could additionally focus on improving oral health, nutrition, sleep quality, and mental health to improve function and health.

This study has several limitations, including limitations of the sample. There is no register of PEH in the host city; thus, there was no way of identifying a comprehensive reference population. Those approached to participate reflected only those people known to staff at the host organisation or its networks. The invitation may not have reached PEH who did not frequent these organisations or who did not regularly visit the inner city. Staff at the host and satellite organisations were well known to participants, and they were responsible for recruitment. This raises issues of potential selection bias, hand picking, or coercion and creates the potential for power imbalances between researchers and subjects. The self-selected volunteer nature of the sample raises concerns over motivations for participation and whether those who declined to participate differed from those who did. As homelessness is a concern for researchers from many disciplines, it may be possible that those who declined to participate did so from research fatigue, and/or mistrust of researchers’ motives. Additionally, we did not collect data on the length of time the participant had been homeless, and this may have a bearing on the results. There was the potential, in the self-reported data (nutrition, hearing-related disability, frailty, anxiety and depression, sleep quality) and assessor-prompted assessment (cognition and memory), for latent Hawthorne effects to have occurred, where individuals modify or inflate behaviours because they know they are being observed [91]. Thus, the consistently poorer findings in this sample against the comparison population data may have occurred because participants manipulated their responses to suit their perceptions of the purpose of data collection. Many participants were familiar with the K10 instrument [92], as they reported that it was regularly used during their health checks. Many could quote questions before they were asked, and there were unsolicited comments that participants were better or worse compared with the last administration.

Lastly, we did not explore instrumental activities of daily living, such as financial and medication management and the economic disadvantage associated with this. The reasons for these consistently significantly poorer findings for the PEH may, however, reflect the reality of homelessness and its contexts in a large city. Poor sleep quality, poor mental health, and poor cognition and memory may reflect the lack of safety/security in lived environments (for instance shared accommodation), unstructured daily routines, and/or not being in employment [78,93]. Poor nutrition, being ‘at-risk’ of a high BMI, waist circumference, and waist-to-hip ratio may reflect lack of finances for healthy food, irregular meals, and food insecurity due to a lack of opportunity for meal preparation and safe food storage [78].

## 5. Conclusions

Homelessness is a complex and potentially reversible social circumstance and is known to be associated with accelerated ageing and incipient frailty. This study reports significant health deficits which are likely to contribute to accelerated ageing. This work with an inner-city cohort living in a high-income country has shown a path to providing a relatively easily implemented group of health assessments to help unpack and therefore subsequently respond to prioritised areas of functional decline and underlying causes for accelerated ageing. The outcomes of this study can be used to inform the prioritisation and development of community-based health services for oral health, sleep quality, and mental health specific to the needs of PEH.

## Figures and Tables

**Table 1 ijerph-22-00135-t001:** Health assessments, by gender and age group differences (results showing significantly more likely are in bold).

	Overall	Male		Female		Less than 55 Years		55 Years and Over	
	*n*/N (%, 95% CI)	*n*	% (CI, 95% CI)	*n*	% (CI, 95% CI)	*n*	% (CI, 95% CI)	*n*	% (CI, 95% CI)
Physical health (objective measurement)									
Over recommended waist-to-hip ratio ^¥¥^	33/46 (71.7, 56.5–84.0)	22	**84.6 (65.1–95.6)**	11	55.0 (13.5–76.9)	19	65. 5 (45.7–82.1)	14	**82.4 (56.6–96.2)**
Ear health (wax, wax occlusion, pus/discharge)	16/42 (38.1, 23.6–54.4)	10	43.5 (23.2–65.5)	6	31.6 (12.6–56.5)	7	26.9 (11.6–47.8)	9	**55.6 (30.0–80.2)**
Unknown hearing problem ^¥¥^	17/49 (34.7, 21.7–49.6)	13	**48.1 (28.7–68.0)**	4	18.2 (5.2–40.3)	8	25.8 (11.8–44.6)	9	**50.0 (26.0–74.0)**
Other physical health (self-reported)									
FMS (at risk for 3 or more movements) ^¥^	27/43 (62.8, 46.7–77.0)	11	44.0 (24.4–65.1)	5	27.8 (9.7–53.5)	9	31.0 (15.3–50.8)	7	**50.0 (23.0–77.0)**
One or more of OHIP-14 domains affected ^¥^ ǂ	26/44 (59.1, 43.2–73.7)	14	56.0 (34.9–75.6)	12	63.2 (38.5–83.7)	13	50.0 (29.9–70.1)	13	**72.2 (46.5–90.3)**
Poor sleep quality (PSQI) ^¥^	27/48 (56.2, 41.2–70.5)	15	53.6 (33.9–72.5)	12	60.0 (36.0–80.9)	19	**65.5 (45.7–82.1)**	8	42.1 (20.2–66.5)
Poor appetite in past week ^¥^	26/52 (50.0, 35.9–64.2)	15	51.7 (32.5–70.5)	11	47.8 (26.8–69.4)	14	45.2 (27.3–64.0)	12	**57.1 (34.0–78.2)**
Obstructive/restrictive/mixed-pattern airflow * ^¥^	21/46 (45.6, 30.9–61.0)	11	44.0 (24.4–65.1)	10	47.6 (25.7–70.2)	12	41.4 (23.5–61.1)	9	**52.9 (27.8–77.0)**
Frailty—vulnerable, mildly/moderately frail ^¥^	23/52 (44.2, 30.5–58.7)	14	46.7 (28.3–65.7)	9	40.9 (20.7–63.6)	11	35.5 (19.2–54.6)	12	**57.1 (34.0–78.2)**
Fall/near fall in past 6 months ^¥^	22/51 (43.1, 29.3–57.8)	11	39.3 (21.5–59.4)	11	47.8 (26.8–69.4)	12	38.7 (21.8–57.8)	10	**50.0 (27.2–72.8)**
Food insecurity in past week ^¥¥^	21/50 (42.0, 28.2–56.8)	13	**46.4 (27.5–66.1)**	8	36.3 (17.2–59.3)	11	36.7 (19.9–56.1)	10	**50.0 (27.2–72.8)**
Oral frailty—less than 20 functioning teeth ^¥^	12/37 (32.4, 18.0–49.8)	7	33.3 (14.6–57.0)	5	31.3 (11.0–58.7)	3	15.0 (3.2–37.9)	9	**52.9 (27.8–77.0)**
Psychological health (self-reported)									
Psychological distress (K10) ^¥¥^	26/47 (55.3, 40.1–69.8)	11	40.7 (22.4–61.2)	15	**75.0 (50.9–91.3)**	18	**62.1 (42.3–79.3)**	8	44.4 (21.5–69.2)
Mental health condition ^¥^	21/53 (39.6, 26.5–54.0)	12	40.0 (22.6–59.4)	9	39.1 (19.7–61.4)	17	**54.8 (36.0–72.7)**	4	18.2 (5.2–40.3)
Health service use (self-reported)									
Not undertaken vaccination **—past 5 years ^¥^	27/53 (50.9, 36.8–64.9)	14	46.7 (28.3–65.7)	13	56.5 (34.5–76.8)	21	**67.7 (48.6–83.3)**	6	27.3 (10.7–50.2)
Emergency presentation—past year *** ^¥^	17/53 (32.1, 19.9–46.3)	9	30.0 (14.7–49.4)	8	34.8 (16.4–57.3)	8	25.8 (11.8–44.6)	9	**40.9 (20.7–63.6)**

^¥^ *p* < 0.05 for age or gender; ^¥¥^
*p* < 0.05 for both age and gender; * lung function spirometry testing; ** once or more in last 12 months; *** flu, meningococcal, pneumonia, shingles; ǂ numbers reporting fairly/very often/always; FMS: functional movement scale; K10: Kessler Psychological Distress Scale; OHIP-14: Oral Health Impact Profile-14; PSQI: Pittsburgh Sleep Quality Index.

## Data Availability

Datasets used and/or analysed during the current study are available from the corresponding author on reasonable request.

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
