# Peer review of "Health Deficits Among People Experiencing Homelessness in an Australian Capital City: An Observational Study"

_ijerph, 2025, doi:10.3390/ijerph22020135_

Round 1
Reviewer 1 Report
Comments and Suggestions for Authors
Thankyou for the opportunity to review this paper. It was very professionally written and presented and makes an important contribution to scholarship.
I have a series of comments to make, summarised in the below bullet points:
- The literature reviewed as part of this study was sound. It was disappointing that more Australian research was not referred to in your study. For example, Professor Cameron Parsell's work on the interface of healthcare and PEH and supportive housing would have been very useful and appropriate.
- The study claims to be based on the AIHW definition of homelessness. In fact, the AIHW use the Australian Bureau of Statistics definition homelessness. This could have been explained in your report. Also, the difference between experiencing homelessness and risk of homelessness could have been explained here.
- The study refers to the service Common Ground in a number of places. It would have been good to outline what type of specialist homelessness service this Common Ground actually is? If the Common Ground mentioned is like other Common Ground services in Australia and overseas, it is actually a Supportive housing approach, not a supported accommodation model. The difference between such service models could have been explained. For instance, a person using a Supported accommodation is still counted as experiencing homelessness whereas this is not the case when a person is housing in a supportive housing model. (Lines 92 in participant recruitment and lines 384-388).
- Following on from the above point, Common Ground services are typically associated for people who have experienced chronic homelessness and rough sleeping. This context would have also been useful to outline in your report, especially as it relates to the limitation you mention about duration of homelessness not actually considered in this study.
- The three page table with quantitative data is too large. and I would recommend finding a way for this table to be more succinct.
- Some brief connection to the housing and homelessness policy settings and environment in Australia would have been useful.
Author Response
Thank you for useful review. We have taken onboard all of your concerns and adapted the manuscript accordingly. Please see attached word document for specific replies to your review.

Reviewer 2 Report
Comments and Suggestions for Authors
Thank you for the opportunity to review this important work. The paper has a very strong background and literature review, methods, results, and discussion. The only issue I wondered about was the repeated reference to a previous paper. I am not sure what is best practice/convention in this journal related to this, but I personally would like to know at least a minimal amount of the information without having to go find a previous article. Otherwise, I strongly support this article and its publication in this journal. Thank you for doing this very important work.
Author Response

(The authors gave the same response as above.)

Reviewer 3 Report
Comments and Suggestions for Authors
see attached file

Author Response

(The authors gave the same response as above.)
